# Analysis of Road Traffic Accidents in Turkey between 2013 and 2017

**DOI:** 10.3390/medicina55100679

**Published:** 2019-10-09

**Authors:** Ali Kemal Erenler, Burak Gümüş

**Affiliations:** 1Emergency Medicine, Department of Emergency Medicine, School of Medicine, Hitit University, Çorum 19000, Turkey; 2Forensic Sciences, Department of Forensic Sciences, School of Medicine, Hitit University, Çorum 19000, Turkey; eti19030@hotmail.com

**Keywords:** Road traffic injury, trauma, mortality

## Abstract

*Background and objectives:* Road traffic accident (RTAs) is one of the top ten leading causes of death worldwide and its incidence is higher in developing countries. In this study, our aim was to determine the characteristics of RTAs in Turkey and make recommendations to reduce mortality and morbidity related to RTAs. *Material and Methods:* We obtained our data, which cover the years 2013 to 2017, from the database accessible at the official website of the Turkish Statistical Institute, which permits the use of its data for research purposes. The chi-square test was used for statistical analysis, and the percentage distribution and odds ratios were calculated. *Results:* In the study period, a total of 697,957 RTAs occurred in Turkey. A total of 1,168,121 individuals have been wounded and 3534 of them have lost their lives. The majority of RTAs occurred on weekends and in summer months. Male individuals are more likely to be exposed to death and injuries related to accidents. When the vehicle type is considered, motorcycle drivers are under more risk for RTAs. RTAs are more likely to occur in rural areas. *Conclusion:* Male individuals and motorcyclists are under a great risk for RTAs. Strict laws are mandatory in order to reduce morbidity and mortality related to RTAs. Additionally, educational efforts must focus on two-wheelers and tractor drivers, particularly in developing countries.

## 1. Introduction

An accident is defined as “an unfortunate incident that happens unexpectedly and unintentionally, typically resulting in damage or injury” [1]. Accidents occurring on the road, involving pedestrians and/or vehicles, are defined as road traffic accidents (RTAs) [2]. RTA is one of the top ten leading causes of death worldwide and its incidence is higher in developing countries [3]. It is also known that RTA is the leading cause of death for young people aged 15–29 years [4]. Accidents are influenced by a combination of human, road and environmental factors [5]. An increase in the related burden has been observed worldwide and pedestrians, cyclists, and two-wheeled motorcycle riders are the most vulnerable populations that contribute to the observed mortality [6].

To the best of our knowledge, there are no published studies evaluating regional premature mortality due to traffic accidents in Turkey. Our aim in this study is to investigate characteristics of RTAs in Turkey and make recommendations for prevention from morbidity and mortality.

## 2. Material and Methods

We obtained our data, which cover the years 2013 to 2017, from the database accessible at the official website of the Turkish Statistical Institute, which permits the use of its data for research purposes. The data were evaluated by using the Statistical Package for the Social Sciences (SPSS) 10.0 program. The chi-square test was used for statistical analysis, and the percentage distribution and odds ratios were calculated.

## 3. Results

In a 5-year-period, a total of 697,957 RTAs occurred in Turkey. As a result of these accidents, a total of 1,168,121 individuals have been wounded and 3534 of them have lost their lives.

When the monthly distribution of these RTAs was investigated, it was determined that the frequency of RTAs rose in summer months when compared to winter months. Accordingly, injuries and death also occurred more frequently in summer months. On weekends (Saturday and Sunday), the frequency of RTAs tends to rise. The proportion of RTAs resulting in death was found to be higher at night and twilight hours. Table 1 summarizes the distribution of number of accidents, deaths and injuries according to months, days and day hours.

Death due to RTAs occurred more frequently in males when compared to females. When driver deaths are considered, males have lost their lives 60-fold more. When pedestrian deaths are investigated, males have died 2.5-fold more than females. Also, injuries related to RTAs are more commonly seen in males. Males are also exposed to driver injuries 14-fold more frequently than females. When passengers and pedestrians are considered, significant differences could not be obtained between genders. Comparison of RTAs according to gender is summarized in Table 2.

When the vehicle type involved in the accident is investigated, death and injury rates were low in buses. When compared according to numbers of vehicles, the death rate was higher in tractor and injury rate was higher in motorcycle accidents. Vehicle types and outcomes of RTAs are summarized in Table 3.

According to the number of drivers, while driver death rate tends to decrease over the years, a significant difference could not be obtained in injury rates. SeeTable 4 for details.

The majority of RTAs and injuries occurred in residential areas. However, death related to RTAs commonly occurred outside of residential areas. Table 5 summarizes the location of RTAs.

## 4. Discussion

Accidents occurring on the road, involving pedestrians and/or vehicles, are defined as RTAs [2]. Globally, RTA is the leading cause of injury-related deaths [7]. According to the reports of World Health Organization (WHO), car accidents will be the fifth leading cause of death in the world by 2030 [8]. In our study, we determined that 3534 individuals have lost their lives in RTAs in a 5-year period. In a study by Tekpa et al., 217 of 1283 victims of RTAs have died, revealing a lethality rate of 16.9% in a 12-month period [9]. In another study involving 50 states of the United States (US), it was reported that a total of 1,220,610 deaths attributable to traffic crashes occurred between 1985 and 2014 [10]. It is known that number of deaths in RTAs and mortality rate are closely related to socio-economical status of countries. In a study in Kosovo, a slight decrease in the mortality rate of 0.1% and lethality rate of 0.1% each year was determined, whereas there was an increase of 21.5% for traumatism rate for each year [4]. Improving the healthcare system plays an essential role in reducing mortality rate.

It is well-known in the literature that RTAs tend to increase in summer months [11]. Accordingly in our study, both RTAs and deaths related to RTAs occurred more frequently in summer months. In summer months, individuals travel more with vehicles and participate in outdoor activities more frequently, thus the number of RTAs increase.

More accidents occur at night than during the day [2]. In a study, it was determined that the majority of RTAs happened between midnight to 6 A.M. [12]. Accordingly in our study, the majority of the incidents happened in night hours in weekends. Contrarily, in another study, RTAs were reported to occur mostly during Monday and in the afternoon rush hours between 14.00–18.00 [4].

In our study, male individuals are found to be exposed to RTAs more frequently than females. According to the WHO report, more than 75.0% of the road traffic deaths happened in males [13]. In a report in a 5-year period, of the 579 RTA casualties examined, 72% were males, 28% females [14]. In another study with a total of 5298 patients, 87.3% were men [15]. Moreover, in a report, the mortality rates due to RTA were four times higher in men [16]. The reason for this result may be the fact that men tend to drive faster and more carelessly when compared to females. Also, drinking and driving may be more common among men.

Our results revealed that the lowest injury and death rates were obtained in buses. This result underlines the importance of public transport. Accordingly, in a study by Fernando et al., individuals involved in RTAs were passengers (44%), drivers (32%), and pedestrians (20%). It was also reported that of the 440 vehicle occupants, 37% were on motorcycles, 28% in three wheelers, 13% in dual purpose vehicles and 11% in buses [14]. In a report, it was stated that passengers in cargo trucks and motorcyclists were the most exposed to fatal accidents [9].Accordingly, in another report, the majority of the patients were injured while riding a motorcycle or scooter [15]. For instance, in Brazil, in a 15-year period, the mortality rate is 7.5 times higher in motorcycle riders and 3.4 times higher in automobile riders [16]. Our study also revealed that the death rate was higher in motorcycle and tractor drivers. Helmet use may decrease risk of head trauma and thus reduce morbidity and mortality in patients with motorcycle accidents [17].Implementation of strict license laws and continuous education for motorcyclists are highly recommended.

Over 50% of crashes occur on urban roads (4). In a report of a total of 56,966 RTAs, it was stated that the majority of RTAs occurred in urban areas. It was also reported that the risk of a fatal accident was 6.8-fold higher in rural areas than in urban areas [18]. In concordance with the literature, our results revealed that majority of the RTAs resulting with death occur in urban areas. As a common problem of developing countries; inadequate road infrastructure, high speed and disobeying traffic rules in urban areas may increase RTA frequency in such areas [4].

## 5. Conclusions

RTAs more commonly occur on weekends and in summer months when people are more active and participate in outdoor activities frequently. Males are under more risk when compared to females. In respect to vehicle type, motorcycle and tractor drivers are under a greater risk when compared to other vehicle drivers. Public transport may be a solution for increasing rates of RTAs. For individual drivers, laws limiting the number of passengers carried, installation of side doors, mandatory use of seatbelts in three wheelers, and protective clothing for motorcyclists are recommended (14). Since roads are common use areas, drivers must be forced to comply with the rules and law-makers must focus on education of drivers and the public.

## Figures and Tables

**Table 1 medicina-55-00679-t001:** Distribution of number of accidents, deaths and injuries according to months, days and day hours.

	Accidentsn (%)		Deaths n (%)		Injuries n (%)	
Monthly Distribution						
Total	697,957	100	3534	100	1,168,121	100
January	42,304	6.06	189	5.35	71,146	6.09
February	40,601	5.82	172	4.87	66,903	5.73
March	48,902	7.01	228	6.45	77,936	6.67
April	55,272	7.92	258	7.30	87,863	7.52
May	62,043	8.89	326	9.22	101,627	8.70
June	63,946	9.16	341	9.65	107,343	9.19
July	72,160	10.34	427	12.08	128,128	10.97
August	75,224	10.78	413	11.69	133,619	11.44
September	68,153	9.76	371	10.50	116,364	9.96
October	63,406	9.08	300	8.49	105,573	9.04
November	56,437	8.09	279	7.89	90,489	7.75
December	49,509	7.09	230	6.51	81,130	6.95
Daily Distribution						
Total	697,615	100	14,236	100	1,164,083	100
Monday	100,441	14.4	2514	17.7	163,873	14.1
Tuesday	95,783	13.7	2460	17.3	154,332	13.3
Wednesday	96,062	13.8	2366	16.6	154,435	13.3
Thursday	95,211	13.6	2385	16.8	154,210	13.2
Friday	103,548	14.8	2471	17.4	169,302	14.5
Saturday	104,471	15.0	2911	20.4	179,665	15.4
Sunday	102,099	14.6	2960	20.8	188,266	16.2
Hourly Distribution						
Total	880,626	100.00	18,067	100.00	1,468,504	100.00
Daylight	589,000	66.88	10,512	58.18	965,590	65.75
Night	266,915	30.31	6843	37.88	460,090	31.33
Twilight	24,711	2.81	712	3.94	42,824	2.92

**Table 2 medicina-55-00679-t002:** Comparison of drivers, passengers and pedestrians according to gender.

		Total	%	Driver	%	Passenger	%	Pedestrian	%
Death	Total	18,067	100,00	7896	100.00	7168	100.00	3003	100.00
	Male *	13,937	77.14	7769	98.39	4034	56.28	2134	71.06
	Female	4130	22.86	127	1.63	3134	43.72	869	28.94
Injury	Total	1.468,504	100.00	617,930	100.00	680,628	100.00	169,946	100.00
	Male *	1,022,671	69.64	578,385	93.60	348,491	51.20	95,795	56.37
	Female	445,833	30.36	39,545	6.40	332,137	48.80	74,151	43.63

* Statistically significant.

**Table 3 medicina-55-00679-t003:** Comparison of outcomes of drivers according to the type of vehicle.

	Number	Drivers Died (n)	%	Drivers Injured (n)	%
Total	1,396,979	7896	0.6	617,930	44.2
Automobile	719,615	3396	0.5	269,699	37.5
Van	220,705	897	0.4	72,160	32.7
Motorcycle	221,404	1549	0.7	189,589	85.6
Tractor	15,440	770	5.0	6474	41.9
Truck	42,538	410	1.0	13,275	31.2
Minibus	43,827	108	0.2	8316	19.0
Bus	34,197	71	0.2	3435	10.0
Tow truck	32,802	357	1.1	10,636	32.4
Others	66,451	358	0.5	44,346	66.7

**Table 4 medicina-55-00679-t004:** Death and injury rates among drivers.

	Total Number of Drivers	Drivers Died (n)	%	Drivers Injured (n)	%
2013	24,778,712	1577	0.00636	113,345	0.4574
2014	25,972,519	1506	0.00580	118,196	0.4551
2015	27,489,150	1658	0.00603	128,036	0.4658
2016	28,223,393	1574	0.00558	129,681	0.4595
2017	28,181,830	1581	0.00561	128,672	0.4566

**Table 5 medicina-55-00679-t005:** Number of accidents, death and injuries according to the scene of the accidents.

	Total (n)	Residential Areas (n)	%	Rural Areas (n)	%
Accidents	1,865,563	1,341,033	71.9	524,530	28.1
Death	56,829	19,983	35.2	36,846	64.8
Injury	3,220,661	2,054,219	63.8	1,166,442	36.2

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
