# Peer review of "Analysis of Road Traffic Accidents in Turkey between 2013 and 2017"

_medicina, 2019, doi:10.3390/medicina55100679_

Round 1

Reviewer 1 Report

The paper aims at analyzing Road traffic accidents (RTA) in Turkey based on a dataset that covers years between 2013 and 2017 by identifying the characteristics of RTAs and providing the related findings.
The subject of the paper is interesting. However, authors should handle many issues within the paper:

1. The paper needs an extensive grammatical correction. There are several mistakes hint that the paper didn’t take enough attention regarding the language. Following are a few of them:

traffic accidents (RTAs) is > traffic accidents (RTAs) are occured  > occurred  investigat characteristics of > investigate the characteristics of pedesterian > pedestrian detah > death etc

2. Title: there are additional blanks “between 2013 and 2017”

3. The abstract needs to be rewritten to show what has been done and what the paper offers. Otherwise, the current abstract looks like an outline of the paper (Introduction: .., Material and Methods:.., etc).

4. The paper is missing a related work section. 

5. What is the real contribution or the added value of this paper? Compared for example with the following papers:

ANALYSIS OF ROAD TRAFFIC ACCIDENTS IN ANTALYA PROVINCE (TURKEY) USING GEOGRAPHICAL INFORMATION SYSTEMS An analysis of road traffic accidents in Turkey using logit models

I believe that the paper can be improved significantly and become a good paper in terms of discussion, structure, and grammar.

Reviewer 2 Report

I have read this paper carefully and I think the authors have made a simple but relevant study about a topic that is important and relevant worldwide.

I have only some minor observations.  

Last paragraph of Results: check misspelling error: “detah related to RTAs”. I guess the authors meant “death related to RTAs”. Check spacing between word, there are several errors. For example:

Last paragraph of Results.  “Table 5summarizes”.

First paragraph of Discussion. “In another studyinvolving”

Second paragraph of Discussion. deaths related toRTAs.

Second paragraph of Discussion. “number of RTAsincrease”.

And the list of this type of mistake continue.

Round 2

Reviewer 1 Report

The authors have responded to my concerns and made necessary changes to the manuscript.

Best regards

Reviewer 2 Report

I have reviewed this new version, and I have no more comments or observations. The authors have provided enough details and their replies are good enough.